# Preoperative and perioperative factors that predict endothelial cell loss 1 year after uncomplicated Descemet membrane endothelial keratoplasty

Jules Leterrier[1], Lucas Mastrangelo[1], Christophe Goetz[2], Yinka Zevering[2], Jean-Marc Perone[1]*

**1** Ophthalmology Department, Mercy Hospital, Regional Hospital Center of Metz-Thionville, Metz, Grand-Est, France, **2** Clinical Support Unit, Mercy Hospital, Regional Hospital Center of Metz-Thionville, Metz, Grand-Est, France

* jean-marc.perone@chr-metz-thionville.fr

## Abstract

### Purpose

To identify pre/perioperative variables that shape endothelial cell loss (ECL) after uncomplicated Descemet membrane endothelial keratoplasty (DMEK).

### Methods

This retrospective study included all consecutive patients with Fuchs endothelial corneal dystrophy who underwent DMEK surgery without perioperative or postoperative complications in 2015–2023 and were followed for 12 months. To identify covariates that predicted 12-month ECL, primary hierarchical multivariable analysis was conducted with 12 variables: patient age and sex; donor age; preoperative axial length, visual acuity, central corneal thickness, and graft endothelial cell density; endotamponade with sulfur hexafluoride (SF6) or air; triple-DMEK or pseudophakic-DMEK; operative time; graft marking; and rebubbling.

### Results

137 eyes (112 patients) were included. Multivariable analysis showed that SF6 predicted $13.6 \pm 3.4\%$ greater ECL *vs*. air ($p < 0.0001$) and accounted for 10% of total ECL variation. Longer operative time and multiple ($\geq 2$) rebubbling also predicted $0.4 \pm 0.7\%$ ($p = 0.046$) and $11.7 \pm 5.1\%$ ($p = 0.02$) higher ECL, respectively. SF6 significantly reduced rebubbling on univariable analysis (13% *vs*. 41% for air, $p = 0.01$).

**Data availability statement:** The datasets generated during and/or analyzed during the current study are not publicly available according to French Law No. 2018-493 of June 20, 2018 on the protection of personal data (The General Data Protection Regulation (Regulation (EU) 2016/679) (GDPR: article 9) but are available from the Clinical Research Support Platform (Plateforme d'Appui à la Recherche Clinique [PARC]) of the Regional Central Hospital (CHR) of Metz-Thionville on reasonable request (email: projetrechercheclinique@chr-metz-thionville.fr, tel: +33 3 87 17 98 82). All nonarchived data is subject to daily backups while all archived data is subject to duplicate storage at two different sites. This data processing is compliant with a baseline reference methodology (MR001) to which the CHR MetzThionville signed a compliance commitment on October 8, 2018.

**Funding:** The author(s) received no specific funding for this work.

**Competing interests:** The authors have declared that no competing interests exist.

## Conclusions

SF6 use for endotamponade may increase ECL after DMEK. There is an urgent need for randomized controlled trials that estimate the relative disadvantages (endothelial toxicity) and advantages (less bothersome rebubbling) of SF6.

## Trial registration

ClinicalTrials.gov Identifier: NCT02535819.

## Introduction

Fuchs endothelial corneal dystrophy (FECD) is caused by genetic factors that result in progressive ablation of the endothelial cells that line the cornea and regulate its hydration. This eventually leads to corneal edema, loss of corneal transparency, and blindness [1]. The gold standard treatment for FECD is Descemet membrane endothelial keratoplasty (DMEK) [2,3]: this involves replacing a disc of diseased Descemet membrane and endothelium with a disc of healthy donor tissue, followed by endotamponade with a bubble of air to promote graft adherence. Many studies show that the thin DMEK graft (15 μm) provides better and faster visual recovery, less graft rejection, and fewer refraction changes than penetrating keratoplasty (full-thickness grafts) [4–7] and Descemet stripping automated endothelial keratoplasty (DSAEK) grafts (45–180 μm thick) [8,9]. However, the thinness of the DMEK graft also means that it scrolls forcefully due to the elasticity of the Descemet membrane. This complicates graft preparation and transplantation and induces graft detachment in the days/weeks after surgery in approximately a quarter of cases [9–11]. To salvage the graft, postoperative rebubbling with one or more air tamponades may be needed.

All keratoplasty techniques induce donor corneal endothelial cell loss (ECL). In DMEK, 30–50% of endothelial cells are lost in the first 6 months. Moreover, after stabilization, annual ECL is six-fold higher than in unoperated eyes (~4% vs. 0.6–0.7%/year) [12,13]. Since graft ECL dictates the long-term success and survival of the graft [9,14–16], many studies have sought to determine preoperative and perioperative factors that increase ECL in DMEK and that could be mitigated or used to identify patients at risk of DMEK graft failure. However, most studies examine only one or a few factors, use univariable analyses rather than multivariable analyses, and/or include bilateral eyes without accounting for the non-independence of these samples. These limitations may partly explain the many inconsistences between studies: for example, graft detachment (or its surrogate measure, rebubbling) associates with increased ECL in some studies [11,17–25] but not others [26–28]. Moreover, some preoperative and perioperative factors are not well researched. In particular, a recent change in the DMEK field involves using a 20% mixture of sulfur hexafluoride (SF6) in air instead of air alone for endotamponade because it increases bubble duration and reduces graft detachment [29–32]. While several studies concluded that SF6 use does not increase ECL [26,29,33–38], most employed univariable analyses [33,36,39–41] and the endothelial cell safety of SF6 has never been tested by

randomized clinical trials (RCTs). Another example is marking of the graft with gentian-violet ink: this prevents the surgeon from misplacing the graft endothelium against the stroma, which causes immediate graft failure [31,42,43]. The endothelial cell safety of graft marking has only been assessed by one univariable study [42]. Moreover, while two studies have found that a short axial length associates with greater ECL in DMEK patients [44,45], this was not observed elsewhere [26,27] and the role of ocular anatomy in post-DMEK ECL remains poorly understood.

To address this, we used multivariable analysis to determine the ability of 12 routinely recorded preoperative and perioperative variables to predict ECL at 12 post-DMEK months in FECD cases. The cases all involved DMEK without perioperative or postoperative complications since they are the majority of DMEK patients and we wanted to identify primary ECL-inducing factors that we could potentially ameliorate or avoid, thereby improving DMEK outcomes.

## Methods

### Study design and ethics

This retrospective single-center cohort study was conducted at the Ophthalmology Department of the Metz-Thionville Regional Hospital Center (Grand Est, France). It adhered to the principles of the Declaration of Helsinki, was approved by the Ethics Committee of the French Society of Ophtalmology (IRB No. 00008855), and was registered on Clinicaltrials. gov (NCT05531760). Prior to surgery, all patients were informed that their surgery-related data might be used for research purposes. All provided written consent according to the procedure outlined by the reference methodology MR-004 of the National Commission for Information Technology and Liberties of France (No. 588909 v1). The data were accessed on 2 January, 2025. JMP had access to information that could identify individual participants during or after data collection.

### Patient selection and data collection

The prospectively maintained medical records were searched for all consecutive adult patients (≥18 years) with FECD who underwent uncomplicated DMEK between 1 October 2015 and 1 April 2023 and were followed for at least 12 months. All patients either underwent simultaneous cataract surgery (termed triple-DMEK) or had previously undergone cataract surgery (termed pseudophakic-DMEK). Patients were excluded if they had an indication other than FECD; the operative eye had undergone grafting previously; there were obvious intraoperative or perioperative complications that could affect postoperative endothelial cell density (ECD) (i.e., graft-preparation/dissection difficulties, challenges when inserting and unfolding the graft, and postoperative complications, including passage of a gas bubble into the retroiridal space and development of significant ocular hypertension); the graft failed; and/or the 12-month ECL data were missing. Cases such as eyes with prior glaucoma surgery, vitrectomized eyes, or eyes with implants fixed to the sclera were not included because we conduct DSAEK with such cases. Eyes with well-controlled glaucoma were not excluded.

### Preoperative, perioperative, and postoperative measurements

Best spectacle-corrected visual acuity (BSCVA) was measured before and 6, 12, and 24 months after DMEK in Monoyer scale. The values were converted to logMAR. Central corneal thickness (CCT) was measured before and 6, 12, and 24 months after DMEK with non-contact ultrasonic pachymetry (Tono pachymeter NT-530P; Nidek Co., Gamagori Aichi, Japan). Donor age and preoperative graft ECD were provided by the eye bank: ECD was determined by manual counting under an optic microscope by an eye-bank technician followed by a recount by another technician. Postoperative graft ECD at 6, 12, and 24 months was determined with specular microscopy (NIDEK CEM-530; NIDEK, Tokyo, Japan). Preoperative axial length, anterior chamber depth (ACD), and lens thickness were measured with optical biometry. Automated anterior-segment optical coherence tomography (AS-OCT) (NIDEK with a special module; Nidek Co., Gamagori Aichi, Japan) was conducted 1, 2, and 4 weeks after DMEK to determine graft attachment. The operative time, measured from the initiation of graft dissection to suture placement, was recorded by a nurse using a stopwatch.

## Surgical technique and postoperative care

During preoperative consultations held several weeks prior to DMEK surgery, all patients underwent a lower peripheral iridotomy at the 6 o'clock position with Nd:YAG Laser (Laser ex-Super Q; Ellex Europe, Medical Quantel, Cournon d'Auvergne, France) to prevent pupillary block during and after surgery. All surgeries were performed by the same experienced surgeon (JMP) as described previously [46]. This surgeon had conducted 50 DMEK, ~100 DSAEK, and hundreds of penetrating keratoplasties before the DMEK cases that were included in the present study [47]. General anesthesia was used unless it was contraindicated, in which case locoregional (peribulbar) anesthesia was employed. Unprepared grafts were obtained from the tissue banks in Besançon or Nancy (France). All were preserved in Eurobio organ culture medium at 31°C and had a requested eyebank-determined ECD of >2200 cells/mm$^2$. In the operating room, each graft was initially trephined to an 8 mm-diameter disc with Hanna's micro-trephine (Busin Punch 17200D 8-mm single use; Moria SA, Antony, France) and then dissected under an operating microscope with disposable curved monofilament forceps (Single-Use Tying Forceps Curved 5 mm Platform 17501; Moria SA). During dissection, the graft was marked with "F" or "F*" on its stromal side with gentian-violet ink from a sterile dermographic surgical pen (Devon Skin Marker; Covidien, Mansfield, USA). After staining with Trypan Blue for 1 minute (VisionBlue, 0.5-ml syringe; DORC Dutch Ophthalmic Research Center, Zuidland, Netherlands), the graft was placed into a DORC injectable system (30G Curved Cannula for air injection; DORC).

The surgical procedure began with paracentesis using a 2.2-mm blade (Securityblade BD, Xstar 2.2-mm, 45 degrees, 37822; Beaver-Visitec International, Waltham, USA) and a Worst 15 blade (Ophthalmic Knife 15 degrees; ALCON, Ruel Malmaison, France). After injecting sterile air into the anterior chamber, 9-mm central descemetorhexis was performed with an inverted Sinskey Price hook (Single Use Price Reverse Hook Sim 17302; Moria SA) and an inverted spatula (90th single use Spatula 17303; Moria SA). The main incision was enlarged to 4 mm to facilitate insertion of the graft into the anterior chamber *via* the DORC injectable system. The graft was unfolded and positioned by external maneuvers with the use of two 27-gauge Rycroft cannulas. After positioning, a bubble of sterile air or 20% SF6 in air was injected into the anterior chamber. We used air for endotamponade until September 2020, at which point we switched to SF6 for all new eyes. The anterior chamber was filled, if possible, to 3/4 or 4/5 of its volume to allow free passage of aqueous humor through the inferior iridotomy. The main incision was then sutured with one point of Nylon 10.0 and buried secondarily. If air was used for endotamponade, resorption occurred in 1–3 days. If 20% SF6 was used, resorption took 7–14 days.

In the case of triple-DMEK, phacoemulsification was performed before DMEK with the standard supracapsular "garde à vous" technique [48] using the Stellaris PC (Bausch and Lomb, Aliso Viejo, CA, USA). A Zeiss CT Asphina 409MV intraocular lens was implanted in the capsular bag, with a refractive target of −0.50 D to compensate for the expected postoperative hyperopic shift [49,50].

At the end of DMEK, 1 mL cefuroxime (Aprokam 50 mg; Thea Laboratory, Clermont-Ferrand, France) was injected intracamerally, and 500 mg acetazolamide (Diamox; Sanofi, Gentilly, France) was given intravenously.

After surgery, all patients were placed in the supine position for the first 12 hours. Thereafter, the supine position was maintained as much as possible until the gas bubble resolved. The patients were followed on days 8 and 15 and months 1, 3, 6, 12, and 24. Starting on the day the bubble resolved, all patients were treated 3 times/day for one month with the non-steroidal anti-inflammatory drug Indocollyre 0.1% (Indometacine 0.1%; Bausch and Lombe, Montpellier, France), Maxidrol (a topical antibio-corticosteroid containing dexamethasone, neomycin, and polymyxine B) (ALCON, Rueil Malmaison, France), and an ophthalmic ointment (Vitamin A dulcis; ALLERGAN, Courbevoie, France). In the next month, Indocollyre was stopped, the ointment was continued for three months (once a day), and Maxidrol was replaced with Flucon (NOVARTIS Pharma, Rueil Malmaison, France), a low-dose corticosteroid eye drop. Specifically, Flucon was administered 3 times/day for 2 months, twice/day for the next year, and once/day thereafter for generally 5 years. AS-OCT was conducted to detect graft detachment during the first few months. Since any central graft detachment directly impacts

visual acuity, rebubbling was always conducted in these cases, regardless of the size of the detached area. Rebubbling was also conducted if ≥20–30% of the total graft area had detached. Specifically, in the first 2–3 years of the study period, we rebubbled when ≥20% detachment was observed; later, we only rebubbled when ≥30% of the graft had detached [51,52]. Rebubbling was performed under topical anesthesia (0.4% oxybuprocaïne hydrochloride; Thea Laboratory) with an operating microscope, and always with air: rebubbling with 20% SF6 was never conducted because the prolonged bubble generated by the latter is not needed in these partial engraftment cases. Rebubbling was repeated if necessary. However, if a fourth rebubbling session was needed, or if the corneal edema persisted after 3 months, the graft was considered to have failed and the patient was scheduled for another keratoplasty.

### Collected data

The following variables were retrieved from patient records: patient age and sex; eye laterality; preoperative axial length, ACD, lens thickness, BSCVA, and CCT; graft-donor age and preoperative graft ECD; use of general anesthesia; use of triple-DMEK; graft marking (yes/no); use of SF6 or air for endotamponade (we switched from routine use of air to routine use of SF6 in September 2020); operative time; rebubbling (none/one/≥2); and 6-, 12-, and 24-month BSCVA, CCT, and graft ECD and ECL. ECL was defined as the percent difference in graft ECD 12 months after surgery relative to preoperative graft ECD.

### Sample size

By using the rule of thumb of 10 patients per independent variable, the sample size required for linear regression with 12 independent variables was estimated to be 120 eyes.

### Statistical analysis

The variables that were selected for study were all routinely recorded variables that could potentially shape ECL, as suggested by the literature [26,28,29,38,42,44]. Continuous and categorical data were expressed as median [interquartile range (IQR)] and $n$ (%), respectively. The univariable relationships between 12-month ECL and preoperative and perioperative variables were initially determined with Wilcoxon or Kruskal-Wallis test or Spearman's correlation. Subsequently, to account for the fact that a fifth of the patients underwent bilateral DMEK, the univariable analyses were repeated with generalized linear model (GLM) regression with a random factor on the patient. Explanatory multivariable analysis with GLM regression with a random factor on the patient was then conducted. After collinearity analyses with eigenvectors, all univariable analysis variables except eye laterality, use of general anesthesia, ACD, and lens thickness were included in this multivariable analysis. ACD and lens thickness were excluded because they are well known to correlate with axial length [53–55]. Spearman's correlation analyses were conducted to assess the correlations between the right and left eyes of bilateral-DMEK patients in terms of preoperative CCT and 12-month ECL. All statistical analyses were performed with SAS software (version 9.4, SAS Inst., Cary, NC, USA). The significance threshold was set at 5%.

## Results

During the study period, 193 eyes of 141 patients underwent DMEK. Of these, 56 eyes were excluded because they had indications other than FECD (pseudophakic bullous keratopathy $n=5$, post-uveitis $n=1$), they had undergone graft surgery previously ($n=7$), there were perioperative complications that could affect ECD ($n=28$), the graft failed ($n=11$), or 12-month ECD data were missing ($n=4$). With regard to the perioperative complications, they were obvious difficulties in graft preparation (dissection) ($n=6$), challenges in graft insertion and unfolding ($n=20$), and passage of a gas bubble into the retroiridal space that induced significant ocular hypertension ($n=2$). The remaining 137 eyes of 112 patients were included in the univariable and multivariable analyses (Fig 1).

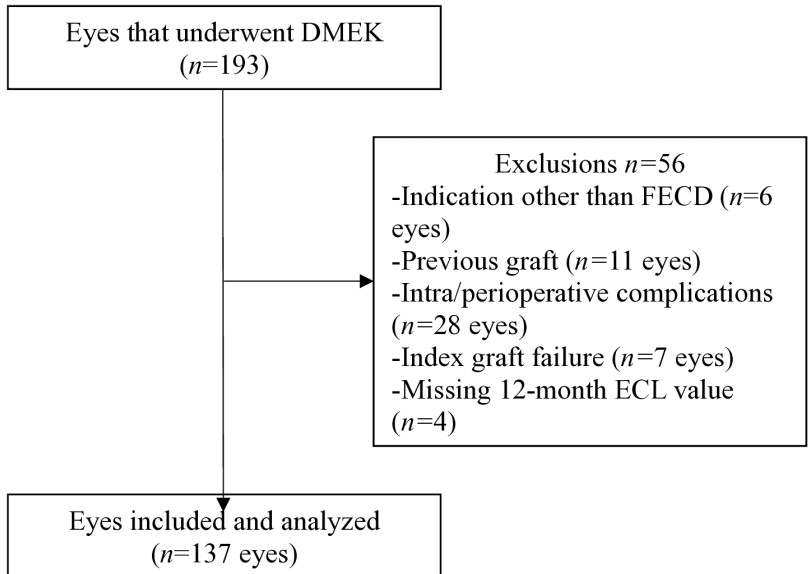

**Fig 1. Patient inclusion flowchart.** The DMEK graft was considered to have failed if a fourth rebubbling session was needed or corneal edema persisted after 3 months. DMEK, Descemet membrane endothelial keratoplasty; ECL, endothelial-cell loss; FECD, Fuchs endothelial corneal dystrophy.

## Baseline clinical and operative characteristics of the cohort and postoperative outcomes

The 112 patients had a median age of 72 years and 61% were women. There were 25 bilateral DMEK cases. Of the 137 eyes, right eyes constituted 55% and median preoperative axial length, ACD, and lens thickness were 23.5, 3.2, and 4.8 mm, respectively. Median preoperative BSCVA and CCT were 0.5 logMAR and 614 μm, respectively. The graft donors had a median age of 75 years and median preoperative-graft ECD was 2560 cells/mm². General anesthesia was used in all cases. Half of the patients underwent triple-DMEK. Graft marking was employed in 77% of cases. SF6 (20%) was used for endotamponade in 34% of cases. Median operative time was 35 minutes. Rebubbling was needed in 31% of cases: one rebubble was required in 23% and ≥2 rebubbles were needed in 8%. Approximately 90% of rebubbling cases were conducted for central detachment. Most rebubbles were conducted 1 week after surgery (S1 Table in S1 File).

Median BSCVA at 6, 12, and 24 postoperative months was 0.17, 0.12, and 0.08 logMAR, respectively. Median CCT at 6, 12, and 24 postoperative months was 540, 538, and 537 μm, respectively. Median graft ECD was 1200, 1000, and 1000 cells/mm² at 6, 12, and 24 postoperative months, respectively. ECL at these timepoints was 53%, 61%, and 61%, respectively (S1 Table in S1 File). Most eyes experienced significant ECL at 12 postoperative months: 60% had an ECL of ≥50%. The distribution of ECL values was asymmetrical (Fig 2.)

## Univariable associations between 12-month ECL and pre/perioperative variables

Our initial univariable analyses, which did not account for the fact that 22% of the patients underwent bilateral DMEK, showed that greater 12-month ECL was associated with high preoperative CCT (r=0.21, *p*=0.01), triple-DMEK (68% *vs*. 56% ECL for pseudophakic-DMEK, *p*=0.005), graft marking (63% *vs*. 55% ECL for no marking, *p*=0.02), SF6 use (72% *vs*. 54% ECL for air, *p*<0.0001), and longer operative time (r=0.25, *p*=0.003). The remaining variables did not associate significantly with 12-month ECL on these univariable analyses (Table 1).

To account for the non-independence of the bilateral eyes in the cohort, we conducted a second round of univariable analyses on all 137 eyes with generalized linear regression with a random factor on the patient. Compared to the

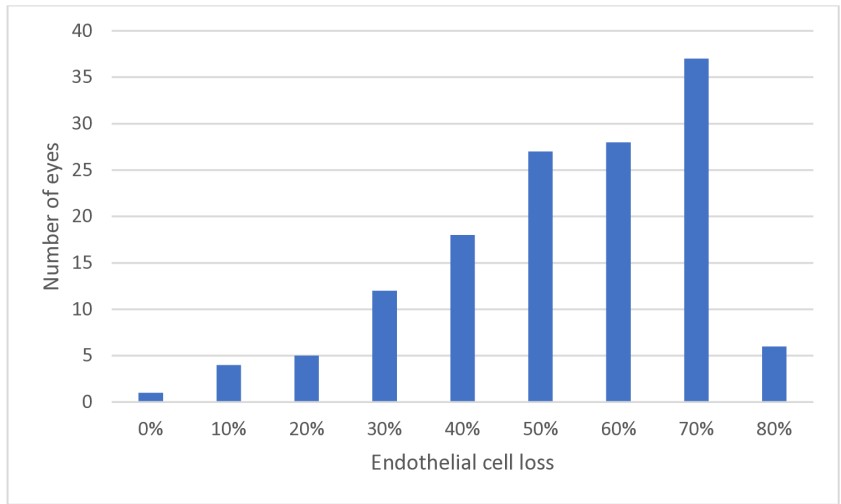

**Fig 2. Distribution of endothelial cell loss values at 12 postoperative months (*n* = 137).**

univariable analyses without this adjustment, similar overall results were obtained except for preoperative CCT, which was no longer associated with 12-month ECL (*p* changed from 0.01 to 0.13) (Table 2). This likely reflects the weak positive correlation between the right and left eye of the bilateral-DMEK patients in terms of preoperative CCT (r = 0.32, 95%CI = −0.12–0.61, *p* = 0.11) (Table 1).

### Multivariable associations between 12-month ECL and pre/perioperative variables

On multivariable analysis that corrected for the bilateral DMEK eyes, SF6 use remained strongly associated with greater ECL (*p* < 0.0001). It accounted for 9.7% of total ECL variance and increased ECL by a median of 13.6 ± 3.4% (Table 3). Longer operative time also predicted greater ECL (*p* = 0.046). It accounted for 2.4% of total ECL variance, and every additional minute of operative time increased ECL by 0.4 ± 0.2% (Table 3). Moreover, relative to no rebubbling, ≥ 2 rebubbles predicted 11.7 ± 5.1% more 12-month ECL (*p* = 0.02) whereas single rebubble was only associated with 1.6 ± 3.4% more ECL (*p* = 0.63) (S1 Fig in S1 File and Table 3). Preoperative CCT, triple-DMEK, and graft marking were no longer associated with high ECL. None of the remaining variable predicted ECL (Table 2).

We next asked whether SF6 use reduced graft detachment. Indeed, GLM regression with patient as a random factor showed that SF6 use was associated with significantly less rebubbling in general: rebubbling was conducted in only 13% of SF6-treated eyes compared to 41% of air-treated eyes (*p* = 0.002). Similarly, only 6% of SF6-treated eyes underwent one rebubble compared to 32% of air-treated eyes. However, this pattern was not obvious for the eyes with ≥ 2 rebubbles: 6% and 9% of SF6- and air-treated eyes underwent ≥ 2 rebubbles, respectively (Table 3).

### Discussion

This retrospective cohort study aimed to identify preoperative and perioperative factors that influenced ECL 12 months after uncomplicated DMEK surgery for FECD. Our primary multivariable analysis showed that use of 20% SF6 for endotamponade strongly predicted 12-month ECL. Longer operative time also predicted 12-month ECL. Moreover, while one rebubble did not significantly increase ECL, ≥ 2 rebubbles did. While univariable analyses suggested that preoperative CCT, axial length, triple-DMEK, and graft marking might associate with 12-month ECL, these relationships vanished on multivariable analysis. Patient age, sex, and preoperative ACD, lens thickness, BSCVA, and graft ECD did not associate with 12-month ECL in any analyses.

**Table 1. Univariable analysis of factors that associate with endothelial cell loss at 12 postoperative months, without and with patient-level correction (*n* = 137).**

| Factor | *n* | % ECL, median (IQR) | Spearman correlation (95% CI) | *p* value without or with patient-level correction | |
|---|---|---|---|---|---|
| | | | | Without, *p*\* | With, *p*\*\* |
| Age, years | 137 | | 0.07 (−0.10–0.23) | 0.42 | 0.76 |
| Sex | | | | 0.76 | 0.81 |
| Men | 53 | 61 (52–72) | | | |
| Women | 84 | 61 (52–72) | | | |
| Donor age, years | 138 | | −0.15 (−0.31–0.02) | 0.08 | 0.16 |
| Preop axial length, mm | 124 | | 0.16 (−0.01–0.33) | 0.07 | 0.19 |
| Preop ACD, mm | 124 | | −0.05 (−0.22–0.13) | 0.62 | 0.97 |
| Preop lens thickness, mm | 69 | | 0.08 (−0.16–0.31) | 0.50 | 0.56 |
| Preop BSCVA, logMAR | 137 | | −0.14 (−0.30–0.03 | 0.11 | 0.21 |
| Preop CCT, µm | 138 | | 0.21 (0.05–0.37) | **0.01** | 0.13 |
| Preop graft ECD, cells/mm² | 138 | | 0.01 (−0.15–0.18) | 0.86 | 0.36 |
| Triple DMEK | | | | **0.005** | **0.03** |
| Yes | 69 | 68 (54–73) | | | |
| No | 68 | 56 (43–69) | | | |
| Graft marking | | | | **0.02** | **0.001** |
| With an F | 106 | 63 (49–72) | | | |
| None | 31 | 55 (43–66) | | | |
| SF6 use | | | | **<0.0001** | **<0.0001** |
| Yes | 47 | 72 (68–77) | | | |
| No | 90 | 54 (41–66) | | | |
| Operative time, min | 138 | | 0.25 (0.09–0.40) | **0.003** | 0.07 |
| Rebubbling | | | | 0.08 | 0.09 |
| None | 94 | 60 (48–72) | | | |
| One | 32 | 58 (43–67) | | | |
| Multiple | 11 | 68 (60–73) | | | |

\*Wilcoxon or Kruskall Wallis test, or Student's test for Spearman correlation vs. zero.

\*\*Generalized linear regression with a random factor on the patient.

ACD, anterior chamber depth; BSCVA, best spectacle-corrected visual acuity; CCT, central corneal thickness; CI, confidence intervals; DMEK, Descemet membrane endothelial keratoplasty; ECD, endothelial cell density; IQR, interquartile range; preop, preoperative; SF6, 20% sulfur hexafluoride.

The demographics of our cohort are similar to those of FECD patients who underwent DMEK in the literature [12]. The visual outcomes were very good: they fell from 0.5 logMAR to 0.08 logMAR. This is similar to that observed in other studies [12,30,52,56,57] and indicates the excellent performance of DMEK in FECD.

## Relationship between SF6 and ECL

Most, albeit not all [58], studies show that use of 20% SF6 for endotamponade significantly reduces DMEK-graft detachment and rebubbling rates [21,26,29,30,33–36,39–41]. Indeed, we also observed that SF6 use reduced rebubbling by ~3-fold from 41% of DMEK cases to 13%. However, we also found that SF6 use strongly predicted greater ECL at 12 postoperative months: compared to air, SF6 increased 12-month ECL by a median of 14%. SF6 use also accounted for 10% of total 12-month ECL variance, making it the strongest predictor of the 12 factors that we tested. These findings

**Table 2. Multivariable analysis of factors that predict endothelial cell loss at 12 months ($n = 137$).**

| Factor | Parameter estimate | Partial $R^2$ | $p^*$ |
|---|---|---|---|
| Female sex | 0.026 ± 0.028 | 0.5% | 0.36 |
| Patient age, per year | 0.001 ± 0.002 | 0.1% | 0.73 |
| Donor age, per year | −0.002 ± 0.001 | 1.8% | 0.08 |
| Preop axial length, per 1 mm | 0.013 ± 0.011 | 0.9% | 0.23 |
| Preop BSCVA, logMAR | −0.042 ± 0.051 | 0.4% | 0.41 |
| Preop CCT, per 100 µm | 0.029 ± 0.022 | 1.1% | 0.19 |
| Preop ECD, per 1000 cells/mm² | 0.044 ± 0.067 | 0.3% | 0.51 |
| Triple-DMEK | 0.012 ± 0.032 | 0.1% | 0.71 |
| Graft marking | 0.007 ± 0.037 | 0.1% | 0.85 |
| SF6 use | 0.136 ± 0.034 | 9.7% | **<0.0001** |
| Operative time, per min | 0.004 ± 0.002 | 2.4% | **0.046** |
| Rebubbling | | 3.1% | 0.08 |
| None | Ref. | | – |
| One | 0.016 ± 0.034 | | 0.63 |
| Multiple | 0.117 ± 0.051 | | **0.02** |

*Generalized linear regression with a random factor on the patient.

BSCVA, best spectacle-corrected visual acuity; CCT, central corneal thickness; DMEK, Descemet membrane endothelial keratoplasty; ECD, endothelial cell density; preop, preoperative; SF6, 20% sulfur hexafluoride.

**Table 3. Association between SF6 and rebubbling ($n = 137$).**

| Rebubbling | SF6<br>$n = 47$ | Air<br>$n = 90$ | $p^*$ |
|---|---|---|---|
| None | 41 (87%) | 53 (59%) | **0.002** |
| One | 3 (6%) | 29 (32%) | |
| Multiple | 3 (6%) | 8 (9%) | |

*Generalized linear modeling with patient as a random factor.

SF6, 20% sulfur hexafluoride.

were unexpected because a meta-analysis of five retrospective studies that conducted univariable analyses [32] and two more recent retrospective studies that performed multivariable analyses [21,26] found that SF6 does not affect post-DMEK ECL. Our findings are concerning because RCTs that compare SF6 to air in terms of post-DMEK ECL have not yet been conducted. There are also some concerns regarding the preclinical studies that suggested SF6 is equally as toxic as air in terms of corneal endothelium histopathology and ECL. These studies involved injecting air or SF6 into rabbit [59,60] or artificial anterior chambers [61], or treating cultured human corneal endothelial cells with air or SF6 for up to 3 days [62]. Rabbit corneal endothelium regenerates, unlike human corneal endothelium, and the *in vitro* models do not fully capture corneal anatomy or physiology. By contrast, Landry *et al.* showed with cats, whose corneal endothelium does not regenerate, that intracameral SF6 injection is significantly more toxic than intracameral air injection, as shown by the additional loss of 132 endothelial cells/mm² and expansion of endothelial-cell area by 25 µm² over 7–10 days [63]. How SF6 induces more corneal endothelial-cell toxicity than air is not clear but Landry *et al.* noted that SF6 generated significantly more anterior-chamber inflammation in the cat eyes, as shown by slit-lamp assessment of anterior-chamber flare and cells: this difference started on day 3–4 after injection and persisted past day 9, which is when the air bubble had cleared and the SF6 bubble was ongoing. Electron microscopy then showed that the inflammation induced by SF6 resulted in the

formation of a loose fibrinous membrane over the corneal endothelium. The membrane contained monocytes, immature fibroblasts, and epithelioid histiocytes and was associated with flatter and more elongated endothelial cells that had disturbed mitochondrial cristae (a sign of cell stress), were sometimes detached from the Descemet membrane, and demonstrated nonspecific protein accumulation. Significantly, the inflamed membrane and stressed endothelial cells were not observed in air-injected eyes [63]. It is possible that the greater SF6-induced inflammation stressed corneal endothelial cells by increasing local reactive-oxygen species (ROS) levels: Sztarbala *et al*. showed that intravitreal SF6 injection elevated ROS in the aqueous humor and that this could potentially induce destructive membrane peroxidation. While Sztarbala *et al*. did not examine the effect of air injection [64], it is likely that ROS levels are lower in air-injected eyes because (i) they demonstrate less inflammation [63] and (ii) the degree of ocular inflammation in anterior ocular diseases correlates positively with local ROS/oxidative stress levels [65].

It should be noted that while the greater proinflammatory effect of SF6 could be due to greater intrinsic toxity of SF6 compared to air, the prolonged presence of SF6 may amplify this toxicity: 20% SF6 bubbles in human eyes last for 7–14 days compared to 1–3 days for air bubbles, and there is a clear relationship between exposure time to air itself and ECL [66]. This bubble prolongation may also promote SF6-induced endothelial harm *via* two other mechanisms. First, like the air bubble, the SF6 bubble may deprive the endothelial cells of access to the aqueous humor and its nutrients and/or hamper endothelial-cell permeability [67,68]. Second, the bubble, or surface-tension phenomena at the gas-fluid interface, may inflict mechanical trauma on the endothelial cells [69,70].

Together with the paucity and limitations of the preclinical and clinical studies on the endothelial safety of 20% SF6 for DMEK endotamponade, our finding that SF6 may in fact be more toxic to the corneal endothelium than air highlights the vital importance of conducting RCTs on this question. Additional studies, including studies with appropriate preclinical models, are warranted.

### Relationship between multiple rebubbles and ECL

Graft detachments are among the most common adverse events after DMEK surgery [71], and most surgeons use rebubbling to salvage the graft [72]. With a few exceptions [26–28], most studies find that rebubbling is linked to more ECL. These studies include one multivariable and many univariable analyses that show any rebubbling associates with more ECL compared to unrebubbled eyes [11,17–19,22,23,25,38,73,74,75]. Several univariable analyses also show that ≥2 rebubbles associate with more ECL graft detachment than one rebubble [17,18,23,74,76]. Similarly, our study showed that ≥2 rebubbles predicted greater 12-month ECL (S1 Fig in S1 File): this is the first time this finding has been confirmed with multivariable analysis. It should be noted that grafts that undergo detachment may already have more ECL than attached grafts: the Melles group showed that when rebubbling of partially detached grafts was rarely conducted (four of 38 [11%] partial-detachment cases underwent rebubbling), the partially detached grafts together nonetheless showed significantly greater ECL than attached grafts [73]. Moreover, ~40% of the postoperative ECL in the first 6 months occurs within 1 day of surgery, before rebubbling is conducted: when Miron *et al*. studied 24 DMEK eyes for which they could obtain early specular microscopy images, they found that the total graft ECD was 24%, 34%, and 60% lower at day 1, week 1, and 6 months, respectively [77]. These findings suggest that surgical damage and/or inherent endothelial weakness of the graft drive rapid early postsurgical ECL that prevents the graft from adhering to the recipient stroma, thus requiring rebubbling. However, it is also possible that rebubbling itself aggravates ECL. This is supported by the fact that air bubbles in the anterior chamber are inherently toxic to corneal endothelial cells (as discussed above). Moreover, Feng *et al*. argued that if rebubbling induces ECL, there should be a dose effect [74]. Indeed, multiple studies, including Feng *et al*. and the present study, found that ECL worsens with increasing numbers of rebubbles [18,23,74,76]. However, it is also possible that multiple rebubbles simply signify more severe surgical damage/endothelial weakness in the graft. Thus, the question of whether rebubbling induces ECL remains unclear. To address it, it is necessary to measure graft ECD longitudinally before

and after rebubbling. However, this approach is limited by the fact that partially detached grafts are edematous, which hampers accurate measurement of ECD by specular microscopy.

Thus, rebubbling might increase ECL. It is also inconvenient for both the patient and the surgeon. Since SF6 could be more toxic than air as an endotamponade but nonetheless significantly reduces graft detachment/rebubbling, RCTs that measure the ECL tradeoffs between the toxic potential of SF6 and its ability to prevent graft detachment/rebubbling are urgently needed.

### Relationship between operative time and ECL

We found that longer operative time predicted greater ECL in our cohort. We also observed this in our previous study on ECL in DMEK cases, and attributed it to difficult surgical procedures such as problematic graft dissection, which correlated with operative time [28] and may have increased ECL. However, such gross surgical difficulties cannot explain the association between longer operative time and ECL in the present study because cases with obviously problematic graft dissection or graft-insertion/unfolding difficulties were excluded. Thus, the association with operative time in our study may reflect other sources of ECL that associate with longer operative time. Further studies that elucidate these sources are warranted.

### Relationships between other variables and ECL

Our study, like most other multivariable analyses [20,26,27,38,75,78,79], did not find that patient age or sex predicted ECL.

A previous univariable analysis observed that a shorter axial length was associated with more ECL, and it was suggested that this may reflect a smaller anterior chamber, which complicates unfolding of the graft [44]. Siggel *et al.* also reported that a very shallow ACD may lead to worse ECL in DMEK [80]. However, our present study and two other multivariable analyses failed to find that axial length predicts ECL in DMEK [26,27]. Preoperative ACD and lens thickness also did not predict ECL on our multivariable analysis.

Our study showed that preoperative visual acuity, CCT, graft donor age, graft ECD, and triple-DMEK did not predict ECL. This has also been observed in three other multivariable analyses for preoperative visual acuity [26,75,79], three multivariate analyses for preoperative CCT [20,26,75], many other studies for graft donor age [27,28,38,75,78,81–83], one multivariable analysis [27] and a univariable analysis of 857 eyes [83] for preoperative ECD, and three multivariable [20,27,28] studies and two meta-analyses [84,85] for triple-DMEK.

Graft marking was significantly associated with ECL on our univariable analysis but not on multivariable analysis. The univariable association likely reflects the fact that we instituted graft marking around the same time as SF6 use for endotamponade [86]. Our multivariable analysis supports the findings of the single (univariable) analysis in the literature, which showed the S-stamp did not increase ECL [42].

### Importance of accounting for bilateral eyes and conducting multivariable analysis

Our initial univariable analysis showed that higher preoperative CCT was associated with worse ECL ($p = 0.01$) but this association disappeared when DMEK-bilaterality was accounted for in univariable analysis. This likely reflects confounding due to right-left eye correlation with regard to preoperative CCT, as shown by us ($n = 25$, r = 0.29, 95%CI = −0.12–0.61, $p = 0.16$) and others ($n = 17212$, r = 0.754, $p < 0.000001$) [87]. Such right-left eye correlation has also been observed for many other ocular variables [87], including 12-month ECL in our study (r = 0.29, 95%CI = −0.08–0.64, $p = 0.16$). Many ocular pathologies are also bilateral [88]. These relationships reflect shared genetic and environmental factors between the eyes. Another example of confounding in bilateral cases is that complications encountered with the first eye could alter treatment decisions for the second eye: for example, if the first eye receiving a DMEK graft experiences unexplained

severe graft detachment, the surgeon may decide to use DSAEK for the second eye instead of DMEK. Thus, surgeon-intrinsic as well as patient-intrinsic factors can significantly confound univariable analyses. These observations illuminate why it is essential to account for bilaterality when conducting statistics on ocular variables, particularly in observational studies (randomization reduces this problem in RCTs). However, 35–92% of studies on eyes that include bilateral eyes do not do this [89,90]. As shown by the present study, this erroneous statistical practice can lead to biased estimates. Specifically, if the two eyes are in different comparison groups, falsely wide confidence intervals and large *p* values can result. By contrast, if the two eyes are in the same comparison group, confidence intervals and *p* values can be erroneously small. Various analytical methods can be used to account for bilaterality, including clustered or nested analysis of variance (ANOVA), ANOVA with right and left eye included as a 'within subject' factor [89], and, as in our case, GLM regression with a random factor on the patient. It should also be noted that to avoid the issue of bilaterality, many studies select only one eye per subject. However, this approach unnecessarily wastes information and decreases statistical power [90].

It is also vital to conduct multivariable analyses because confounding interactions or imbalances between patient, disease, and surgical variables can lead to misleading associations on univariable analysis. For example, as discussed above, graft marking was associated with ECL on univariable analysis in our study, and this association was lost on multivariable analysis. This reflected confounding due to the adoption of SF6 for endotamponade shortly before we started marking the grafts [86]. If we had not conducted the multivariable analysis, we would have erroneously concluded that graft marking is unsafe for endothelial cells. It is also important to carefully consider which variables to include in the multivariable analysis, since this can significantly affect the predictive accuracy of the model [91].

### Study limitations

This study has some limitations. First, its retrospective nature may have led to selection and information bias, despite prospective recording of the data. Second, this study was monocentric. However, all surgeries were performed by the same surgeon, who had ample experience with DMEK and other keratoplasty procedures before the first patient in the current cohort [47]. This limited confounding due to differing surgeon experience and practices. Third, the sample size was relatively small (*n* = 137). Fourth, the predictive accuracy and reliability of multivariable models depend on the independent variables that were included. Consequently, RCTs are needed to verify our observations. Fifth, all patients had FECD and only those who underwent uncomplicated surgeries were selected. Thus, our results may not be generalizable to other endothelial disorders or DMEK surgery that involves perioperative or postoperative complications. Sixth, preoperative ECD has been reported to be overestimated by eye banks [77,92,93]. This could introduce ECL measurement errors. Seventh, we did not examine the roles of other factors that could induce ECL, including metabolic diseases such as diabetes [20,94], graft-storage factors such as duration [78], and patient noncompliance with requirements such as postoperative supine positioning and treatment. Thus, our multivariable analysis did not control for such factors.

### Conclusions

This study suggests that although SF6 endotamponade in DMEK surgery is widely thought to be no more toxic than air, it may in fact increase ECL. However, SF6 also significantly lowers the incidence of rebubbling procedures, which might be harmful to endothelial cells and are bothersome. At present, only a few preclinical and retrospective (mostly univariable) clinical analyses support the endothelial safety of SF6, yet a third of corneal surgeons now routinely use SF6 for endotamponade after DMEK [95]. Our finding highlights the vital importance of conducting RCTs that ascertain the relative advantages and disadvantages of SF6 endotamponade in terms of ECL for DMEK. Moreover, we confirmed with multivariable analysis that multiple rebubbling induces more ECL than one rebubble, although the mechanisms that drive this remain unclear. Further research assessing the effect of rebubbling on ECL and graft failure is needed, possibly with newer methods such as confocal microscopy, which can determine ECD in edematous corneas more accurately than specular microscopy [96].

## Supporting information

**S1 File. One figure (S1 Fig. Endothelial cell loss at 12 months in eyes that underwent no, one, or multiple rebubbles) and one table (S1 Table. Preoperative, perioperative, and postoperative characteristics of the cohort).**
(DOCX)

## Author contributions

**Conceptualization:** Jean-Marc Perone.

**Data curation:** Jules Leterrier, Lucas Mastrangelo, Jean-Marc Perone.

**Formal analysis:** Christophe Goetz.

**Investigation:** Jules Leterrier, Lucas Mastrangelo, Jean-Marc Perone.

**Methodology:** Christophe Goetz, Jean-Marc Perone.

**Project administration:** Jean-Marc Perone.

**Resources:** Jean-Marc Perone.

**Software:** Christophe Goetz.

**Supervision:** Jean-Marc Perone.

**Visualization:** Yinka Zevering.

**Writing – original draft:** Jules Leterrier, Yinka Zevering.

**Writing – review & editing:** Jules Leterrier, Lucas Mastrangelo, Christophe Goetz, Yinka Zevering, Jean-Marc Perone.

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
