## [Decision Letter · Decision Letter 0]

24 Oct 2025

Dear Dr. Perone,

Thank you for submitting your manuscript to PLOS ONE. After careful consideration, we feel that it has merit but does not fully meet PLOS ONE’s publication criteria as it currently stands. Therefore, we invite you to submit a revised version of the manuscript that addresses the points raised during the review process.

We look forward to receiving your revised manuscript.

Kind regards,

Yu-Chi Liu, MD, PhD

Academic Editor

PLOS ONE

Journal Requirements:

Reviewers' comments:

Reviewer's Responses to Questions

**Comments to the Author**

1. Is the manuscript technically sound, and do the data support the conclusions?

Reviewer #1: Partly

Reviewer #2: Partly

2. Has the statistical analysis been performed appropriately and rigorously?

Reviewer #1: No

Reviewer #2: Yes

3. Have the authors made all data underlying the findings in their manuscript fully available?

Reviewer #1: Yes

Reviewer #2: Yes

4. Is the manuscript presented in an intelligible fashion and written in standard English?

Reviewer #1: Yes

Reviewer #2: Yes

Reviewer #1: Authors describe the affecting factors on ECL in FECD patients following DMEK.

The idea is interesting, whereas the article require significant improvement.

The discussion part is quite long. Please shorten the length that is comparable to the introduction part.

The statistics should be evaluated by a specialist.

It seems that the statistics have been performed using mixed factors consisting of preoperative, and postoperative factors.

How to select the type of tamponade? Multiple analysis might lead to different conclusions depending on the evaluating factors. How to justify the selected factors?

The discussion part is quite long. Please shorten the length that is comparable to the introduction part.

Reviewer #2: Comments :

The authors comprehensively evaluated potential risk factors for endothelial cell loss at 1 year after DMEK through extensive multivariate analyses. Although the manuscript is detailed and informative, I would like to highlight several major concerns below.

Title

In the short title, using “Endothelial Cell Loss” instead of “ECL” would be more meaningful and reader-friendly.

Abstract

Line 36 : In the phrase “Fuchs- Endothelial…”, it would be more appropriate to remove the hyphen between the words.

Line 47 : “Younger donor age tended to predict ECL (p=0.08).” The p-value is not below 0.05 and therefore not statistically significant; however, it has been interpreted as significant in the text. Normally, when the p-value exceeds 0.05, further post hoc analyses should not be performed. Nonetheless, a subsequent p-value of 0.02 is also reported. This section should be carefully reviewed for consistency and accuracy.

Introduction :

Line 63-64 : The use of the word “loss” twice in close succession disrupts the flow of the sentence.

Additionally, the Introduction section is rather lengthy; shortening it would improve readability and make the text more concise and engaging.

Methods :

Line 116 : should be ‘adhered’.

Line 202-205 : The postoperative topical steroid regimen has been mentioned; however, it requires further elaboration. It is unclear which specific agent is referred to as a “low-dose steroid.” The tapering strategy should be described more clearly, as ocular hypertension is a well-established postoperative risk following DMEK and is known to negatively affect endothelial cell density. Moreover, the rationale behind the continued use of topical steroids two to three times daily for five years should be clarified. Typically, the dosage is reduced to once daily by the end of the first year, and in some cases, clinicians even discontinue steroids after that period.

Line 210 : Was rebubbling always performed using air? Why was SF6 not utilized? It should be noted that SF6 may be preferred in certain situations where longer intraocular tamponade is desired—such as in eyes with previous glaucoma surgery, vitrectomized eyes, or those with scleral-fixated intraocular lenses.

Line 270-271 : The indications for rebubbling should be described in greater detail, particularly in the Materials and Methods section. According to the general literature, rebubbling is typically recommended when graft detachment exceeds one-third of the graft area. It is also unclear what is meant by “central detachment” in this context. For instance, if there is a 50% inferior detachment, is rebubbling not performed in such cases?

Results :

Line 262-277 : In the Results section, some data are presented in detail both in the text and in Tables 1 and 2. To avoid redundancy, if the data are already provided in the tables, the text should summarize them more concisely—focusing on statistical significance rather than repeating numerical values.

Line 285-288 : Here again, many data points were analyzed with further statistical tests despite p-values not being below 0.05. An explanation for this approach is necessary, as it deviates from standard statistical practice.

Line 315-316 : This part is not appropriate for the Results section; explanations supported by references should be presented in the Discussion section. Additionally, the abbreviation GLM has not been defined earlier in the text and should be clarified.

Line 317 and 327 :It should be ‘SF6 use WAS associated with..’

Line 332 : It should be “was” instead of “is”, as the text generally uses the past tense. Overall, there is inconsistent use of present and past tense throughout the manuscript, and it would benefit from a thorough English language review.

Line 325-349 : The section beginning with “Multivariate analyses exploring relationships between variables in terms of ECL” reads more like a Discussion than a Results section and requires substantial revision. For the entire manuscript, the Results section should focus solely on reporting findings, avoiding interpretations or references to previous studies. Explanations and contextual discussion of the results should be reserved for the Discussion section.

Line 364-366 : “Older donor age promoted ECL when operative time was long and also increased ECL somewhat when SF6 was used.” According to the Abstract, young donor age was listed as a risk factor. This should be checked for consistency with the Results and Discussion sections to ensure that the finding is accurately reported and properly contextualized.

Line 367-375 : Again, although this is part of the Results section, it contains interpretative comments that belong in the Discussion section.

Discussion

Line 479-480 : “To conclude this section, rebubbling is inconvenient for both the patient and the surgeon, and it might increase ECL.” Although it may cause discomfort, a detached graft will result in prolonged corneal edema and delayed visual recovery. Therefore, it can be misleading to conclude that rebubbling should be avoided

Line 534-535 : The p-value here appears somewhat high to be considered “almost significant.”

Line 606 : The use of multivariate analyses undoubtedly adds value to the study. However, it appears that a large number of parameters were assessed. Were all potential variables that could be significant for ECL included in the multivariate analysis, or was the evaluation limited to a smaller set of data chosen based on consistency with the literature?

In addition to the above comments, one of the most important limitations of the study is the lack of evaluation of patients’ preoperative ocular characteristics. Although the patients are described as uncomplicated, no details are provided. For example, lens status, prior vitrectomy, presence of glaucoma, or history of previous glaucoma surgery should be reported. These characteristics should be incorporated into the study results and assessed using appropriate correlation analyses.

The manuscript would also benefit from a comprehensive English language review, particularly regarding the use of articles (e.g., “the”).

Furthermore, the manuscript is very long—specifically, the Discussion section exceeds ten pages. For improved readability, it is recommended to shorten the text, avoid repetition, and ensure that points appropriate for the Discussion are not presented in the Results section.

**Do you want your identity to be public for this peer review?** For information about this choice, including consent withdrawal, please see our Privacy Policy

Reviewer #1: No

Reviewer #2: No

---

## [Author Response · Author response to Decision Letter 1]

21 Nov 2025

Response to Reviewers

General comment to the Academic Editor and Reviewers

During revision, we decided to remove the post-hoc multivariable analysis data to reduce the length and complexity of the manuscript. The data that remain are the originally planned univariable and multivariable analyses. We believe that this better highlights our main finding – that SF6 use may in fact be toxic to the corneal endothelium – and has made the manuscript easier to read. We are grateful to the reviewers for their time and helpful comments.

Journal Requirements

Comment 1:

Reply: The paper was adapted to conform to these style requirements.

Comment 2:

Reply: In France, the dataset cannot be shared on a repository, even when the patients have been anonymized. Legally, we cannot add to or change the original Data Availability statement, which is:

“The datasets generated during and/or analyzed during the current study are not publicly available according to French Law No. 2018-493 of June 20, 2018 on the protection of personal data (The General Data Protection Regulation (Regulation (EU) 2016/679) (GDPR: article 9) but are available from the Clinical Research Support Platform (Plateforme d’Appui à la Recherche Clinique [PARC]) of the Regional Central Hospital (CHR) of Metz-Thionville on reasonable request (email: projetrechercheclinique@chr-metz-thionville.fr, tel: +33 3 87 17 98 82). All nonarchived data is subject to daily backups while all archived data is subject to duplicate storage at two different sites. This data processing is compliant with a baseline reference methodology (MR001) to which the CHR MetzThionville signed a compliance commitment on October 8, 2018.”

Comment 3:

Reply: No previously published works were recommended by the reviewers.

Reviewer 1

Reviewer #1: Authors describe the affecting factors on ECL in FECD patients following DMEK.

The idea is interesting, whereas the article require significant improvement.

Reply: Thank you very much for the time you have taken to review our manuscript, and for your helpful comments. We feel the manuscript is much improved after revision according to reviewer comments.

Please note that during revision, we decided to remove the post-hoc multivariable analysis data to reduce the length and complexity of the manuscript. The data that remain are the originally planned univariable and multivariable analyses. We believe that this better highlights our main finding – that SF6 use may in fact be toxic to the corneal endothelium – and has made the manuscript easier to read.

Comment 1:

The discussion part is quite long. Please shorten the length that is comparable to the introduction part.

Reply: The Discussion was reduced by a third during revison.

Comment 2:

The statistics should be evaluated by a specialist.

Reply: One of the authors, Christophe Goetz, is the head statistician of the hospital.

Comment 3:

It seems that the statistics have been performed using mixed factors consisting of preoperative, and postoperative factors.

Reply: The multivariable analysis was conducted to determine the association between 12-month endothelial cell loss (a postoperative variable) and 12 preoperative and perioperative variables.

Comment 4:

How to select the type of tamponade?

Reply: We routinely used air for the endotamponade for the first 5 years of the study period (October 2015–August 2020). In September 2020, we reviewed the literature and decided to switch to using 20% SF6 for endotamponade, in line with many other ophthalmological units. Thus, for the remaining 2.5 years of the study period (September 2020–April 2023), we routinely used 20% SF6. This point is indicated in the Methods:

“We used air for endotamponade until September 2020, at which point we switched to SF6 for all new eyes.” (Lines 172-173)

Comment 5:

Multiple analysis might lead to different conclusions depending on the evaluating factors. How to justify the selected factors?

Reply: The preoperative and perioperative variables included in our study were all routinely recorded variables that could potentially affect endothelial-cell loss, as indicated by the literature. For example, some studies suggest that patient age, donor age, preoperative axial length, and SF6 use for endotamponade shape, or could potentially shape, ECL (e.g. doi: 10.1371/journal.pone.0264401, doi: 10.1167/iovs.10-6187a, doi: 10.1097/ICO.0000000000001385). However, most studies used univariable analyses, and the few multivariable analyses had a limited range of variables and did not concomitantly assess SF6 use, axial length, and graft marking. This point is indicated in the Introduction:

“many studies have sought to determine preoperative and perioperative factors that increase ECL in DMEK and that could be mitigated or used to identify patients at risk of DMEK graft failure. However, most studies examine only one or a few factors, use univariable analyses rather than multivariable analyses, and/or include bilateral eyes without accounting for the non-independence of these samples... Moreover, some preoperative and perioperative factors are not well researched.” (Lines 73-81)

To address this comment, we added the following text to the Statistics section:

“The variables that were selected for study were all routinely recorded variables that could potentially shape ECL, as suggested by the literature [28,29,38,42,44,53].” (Lines 225-226)

We also emphasized the point that the multivariable analysis results could be limited by variable selection in the Study Limitations section:

“Fourth, the predictive accuracy and reliability of multivariable models depend on the independent variables that were included. Consequently, RCTs are needed to verify our observations. ...Seventh, we did not examine the roles of other factors that could induce ECL, including metabolic diseases such as diabetes [20,96], graft-storage factors such as duration [80], and patient noncompliance with requirements such as postoperative supine positioning and treatment. Thus, our multivariable analysis did not control for such factors.” (Lines 486-496)

Comment 6:

The discussion part is quite long. Please shorten the length that is comparable to the introduction part.

Reply: The Discussion was reduced by 30%.

Reviewer 2

The authors comprehensively evaluated potential risk factors for endothelial cell loss at 1 year after DMEK through extensive multivariate analyses. Although the manuscript is detailed and informative, I would like to highlight several major concerns below.

Reply: Thank you very much for the time you have taken to review our manuscript, and for your helpful comments. We believe the changes requested manuscript have significantly improved the manuscript, including its readability.

Please note that during revision, we decided to remove the post-hoc multivariable analysis data to reduce the length and complexity of the manuscript. The data that remain are the originally planned univariable and multivariable analyses. We believe that this better highlights our main finding – that SF6 use may in fact be toxic to the corneal endothelium – and has made the manuscript easier to read.

Title

Comment 1:

In the short title, using “Endothelial Cell Loss” instead of “ECL” would be more meaningful and reader-friendly.

Reply: To adhere to the character count limit, we changed the short title to “Factors predicting post-DMEK endothelial cell loss” (Line 23).

Abstract

Comment 2:

Line 36 : In the phrase “Fuchs- Endothelial…”, it would be more appropriate to remove the hyphen between the words.

Reply: The hyphen was removed.

Comment 3:

Line 47 : “Younger donor age tended to predict ECL (p=0.08).” The p-value is not below 0.05 and therefore not statistically significant; however, it has been interpreted as significant in the text. Normally, when the p-value exceeds 0.05, further post hoc analyses should not be performed. Nonetheless, a subsequent p-value of 0.02 is also reported. This section should be carefully reviewed for consistency and accuracy.

Reply: During revision, any mention of data that did not achieve statistical significance was deleted. Thus, the revised Abstract no longer mentions results relating to donor age:

“Results: 137 eyes (112 patients) were included. Multivariable analysis showed that SF6 predicted 13.6±3.4% greater ECL vs. air (p<0.0001) and accounted for 10% of total ECL variation. Longer operative time and multiple (≥2) rebubbling also predicted 0.4±0.7% (p=0.046) and 11.7±5.1% (p=0.02) higher ECL, respectively. SF6 significantly reduced rebubbling on univariable analysis (13% vs. 41% for air, p=0.01).” (Lines 44-48)

Introduction

Comment 4:

Line 63-64 : The use of the word “loss” twice in close succession disrupts the flow of the sentence.

Reply: The sentences were rephrased as:

“Fuchs endothelial corneal dystrophy (FECD) is caused by genetic factors that result in progressive ablation of the endothelial cells that line the cornea and regulate its hydration. This eventually leads to corneal edema, loss of corneal transparency, and blindness [1].” (Lines 56-58)

Comment 5:

Additionally, the Introduction section is rather lengthy; shortening it would improve readability and make the text more concise and engaging.

Reply: The Introduction was reduced by 20%.

Methods

Comment 6:

Line 116 : should be ‘adhered’.

Reply: This change was made (Line 103).

Comment 7:

Line 202-205 : The postoperative topical steroid regimen has been mentioned; however, it requires further elaboration. It is unclear which specific agent is referred to as a “low-dose steroid.” The tapering strategy should be described more clearly, as ocular hypertension is a well-established postoperative risk following DMEK and is known to negatively affect endothelial cell density. Moreover, the rationale behind the continued use of topical steroids two to three times daily for five years should be clarified. Typically, the dosage is reduced to once daily by the end of the first year, and in some cases, clinicians even discontinue steroids after that period.

Reply: Thank you for your careful review. We had not described the timing of the postoperative steroid regimen properly. It has been corrected as follows:

“Starting on the day the bubble resolved, all patients were treated 3 times/day for one month with the non-steroidal anti-inflammatory drug Indocollyre 0.1% (Indometacine 0.1%; Bausch and Lombe, Montpellier, France), Maxidrol (a topical antibio-corticosteroid containing dexamethasone, neomycin, and polymyxine B) (ALCON, Rueil Malmaison, France), and an ophthalmic ointment (Vitamin A dulcis; ALLERGAN, Courbevoie, France). In the next month, Indocollyre was stopped, the ointment was continued for three months (once a day), and Maxidrol was replaced with Flucon (NOVARTIS Pharma, Rueil Malmaison, France), a low-dose corticosteroid eye drop. Specifically, Flucon was administered 3 times/day for 2 months, twice/day for the next year, and once/day thereafter for generally 5 years.” (Lines 188-197)

Comment 8:

Line 210 : Was rebubbling always performed using air? Why was SF6 not utilized? It should be noted that SF6 may be preferred in certain situations where longer intraocular tamponade is desired—such as in eyes with previous glaucoma surgery, vitrectomized eyes, or those with scleral-fixated intraocular lenses.

Reply: Yes, rebubbling was always performed with air, not 20% SF6, because the air bubble lasts 1–3 days whereas the SF6 bubble lasts 7–14 days. The prolonged bubble is not needed in rebubble cases because partial engraftment has already occurred. Moreover, this approach minimizes the contact of the graft endothelium with the bubble, which is toxic to the endothelial cells.

We agree that SF6 rebubbling could be useful in the cases you mentioned but in our hospital, we treat such complex cases with DSAEK: it is easier to conduct and much less prone to graft detachment than DMEK.

To address these comments, we added the following texts to the Methods section:

“Rebubbling was performed under topical anesthesia (0.4% oxybuprocaïne hydrochloride; Thea Laboratory) with an operating microscope, and always with air: rebubbling with 20% SF6 was never conducted because the prolonged bubble generated by the latter is not needed in these partial engraftment cases.” (Lines 203-206)

“Cases such as eyes with prior glaucoma surgery, vitrectomized eyes, or eyes with implants fixed to the sclera were not included because we conduct DSAEK with such cases.” (Lines 124-125)

Comment 9:

Line 270-271 : The indications for rebubbling should be described in greater detail, particularly in the Materials and Methods section. According to the general literature, rebubbling is typically recommended when graft detachment exceeds one-third of the graft area. It is also unclear what is meant by “central detachment” in this context. For instance, if there is a 50% inferior detachment, is rebubbling not performed in such cases?

Reply: Central detachment refers to detachment that affects the center of the cornea: since this directly affects visual acuity, central detachment cases always undergo rebubbling, regardless of the size of detachment. It should be noted that the majority (~90%) of rebubbled cases involve central detachment, as stated in our manuscript (Lines 263–264).

With regard to the second indication for rebubbling, namely, total area detached, the threshold used varies from center to center. For example, some centers rebubble when there is any detachment at all (e.g. doi:10.1016/j.ajo.2017.12.014) while others rebubble if detachment exceeds 20% (e.g. doi:10.1177/11206721221146579) or 30% (e.g. doi: 10.1097/ICO.0000000000001872) of the graft surface. We initially rebubbled relatively liberally in our early cases: cases with ≥20% total detachment were rebubbled. As our experience grew, we started to rebubble only when ≥30% detachment was

---

## [Decision Letter · Decision Letter 1]

7 Dec 2025

Preoperative and perioperative factors that predict endothelial cell loss 1 year after uncomplicated Descemet membrane endothelial keratoplasty

PONE-D-25-45991R1

Dear Dr. Perone,

We’re pleased to inform you that your manuscript has been judged scientifically suitable for publication and will be formally accepted for publication once it meets all outstanding technical requirements.

Kind regards,

Yu-Chi Liu, MD, PhD

Academic Editor

PLOS One

Additional Editor Comments (optional):

Reviewers' comments:

Reviewer's Responses to Questions

**Comments to the Author**

Reviewer #1: All comments have been addressed

Reviewer #2: All comments have been addressed

2. Is the manuscript technically sound, and do the data support the conclusions?

Reviewer #1: Yes

Reviewer #2: Yes

3. Has the statistical analysis been performed appropriately and rigorously?

Reviewer #1: Yes

Reviewer #2: Yes

4. Have the authors made all data underlying the findings in their manuscript fully available?

Reviewer #1: Yes

Reviewer #2: No

5. Is the manuscript presented in an intelligible fashion and written in standard English?

Reviewer #1: (No Response)

Reviewer #2: Yes

Reviewer #1: The revision has been performed properly including statistics, methodology, and the structure(the proper length of description).

Reviewer #2: We thank the authors for providing detailed responses to the comments raised during our evaluation. I believe that the manuscript, in its current form, is acceptable for publication.

**Do you want your identity to be public for this peer review?** For information about this choice, including consent withdrawal, please see our Privacy Policy

Reviewer #1: **Yes: ** TAKAHIKO HAYASHI, MD,PhD,MBA

Reviewer #2: No

---

## [Editor Report · Acceptance letter]

PONE-D-25-45991R1

PLOS One

Dear Dr. Perone,

I'm pleased to inform you that your manuscript has been deemed suitable for publication in PLOS One. Congratulations! Your manuscript is now being handed over to our production team.

Kind regards,

on behalf of

A/Prof Yu-Chi Liu

Academic Editor

PLOS One